# Annealing-Dependent Morphotropic Phase Boundary in the BiMg_0.5_Ti_0.5_O_3_–BiZn_0.5_Ti_0.5_O_3_ Perovskite System

**DOI:** 10.3390/ma15196998

**Published:** 2022-10-09

**Authors:** João Pedro V. Cardoso, Vladimir V. Shvartsman, Anatoli V. Pushkarev, Yuriy V. Radyush, Nikolai M. Olekhnovich, Dmitry D. Khalyavin, Erik Čižmár, Alexander Feher, Andrei N. Salak

**Affiliations:** 1Department of Materials and Ceramics Engineering and CICECO—Aveiro Institute of Materials, University of Aveiro, 3810-193 Aveiro, Portugal; 2Institute for Materials Science and CENIDE—Centre for Nanointegration Duisburg-Essen, University of Duisburg-Essen, 45141 Essen, Germany; 3Scientific-Practical Materials Research Centre of NASB, 220072 Minsk, Belarus; 4ISIS Facility, Rutherford Appleton Laboratory, Chilton, Didcot, Oxfordshire OX11 0QX, UK; 5Institute of Physics, Faculty of Science, Pavol Jozef Šafárik University, 041 54 Košice, Slovakia

**Keywords:** lead-free, high-pressure synthesis, conversion polymorphism, X-ray diffraction, piezoresponse force microscopy

## Abstract

The annealing behavior of (1-*x*)BiMg_0.5_Ti_0.5_O_3_–*x*BiZn_0.5_Ti_0.5_O_3_ [(1-*x*)BMT–*x*BZT] perovskite solid solutions synthesized under high pressure was studied in situ via X-ray diffraction and piezoresponse force microscopy. The as prepared ceramics show a morphotropic phase boundary (MPB) between the non-polar orthorhombic and ferroelectric tetragonal states at 75 mol. % BZT. It is shown that annealing above 573 K results in irreversible changes in the phase diagram. Namely, for compositions with 0.2 < *x* < 0.6, the initial orthorhombic phase transforms into a ferroelectric rhombohedral phase. The new MPB between the rhombohedral and tetragonal phases lies at a lower BZT content of 60 mol. %. The phase diagram of the BMT–BZT annealed ceramics is formally analogous to that of the commercial piezoelectric material lead zirconate titanate. This makes the BMT–BZT system promising for the development of environmentally friendly piezoelectric ceramics.

## 1. Introduction

Solid solutions of perovskite (anti)ferroelectric compounds and other multicomponent systems with continuously changeable chemical compositions have been of great interest for many decades. In such materials, there is a coexistence of different structural phases and phase boundaries, in which their dielectric permittivity and electromechanical coefficients achieve very high values. PbZr_1-*x*_Ti*_x_*O_3_ (PZT) with the range of coexistence of several ferroelectric phases (morphotropic phase boundary, MPB) is an outstanding example [1,2,3,4]. It was generally accepted for a long time that MPB in this solid solution system separates the structural phases of rhombohedral (*R*) and tetragonal (*T*) symmetries and locates close to *x* = 0.47 at room temperature. In 1999, based on a precision synchrotron diffraction study, Noheda et al. [5,6] demonstrated that there is a monoclinic phase (*M_A_*), located between *R* and *T*. In 2014, a detailed study using neutron powder diffraction carried out by Zhang et al. [7] has revealed another monoclinic phase (*M_B_*) with a boundary to the *M_A_* structural phase. It was shown that the intermediate monoclinic phase allows continuous change (rotation) of the polarization vector, and the large piezoelectric response of PZT at MPB can be associated with this effect [6]. The motion of the domain walls and the interphase boundaries in the external field was demonstrated to contribute to the induced polarization as well [8,9]. In relaxor ferroelectrics with MPB, such as PbMg_1/3_Nb_2/3_O_3_–PbTiO_3_ and PbZn_1/3_Nb_2/3_O_3_–PbTiO_3_, the movement (flipping and breathing) of numerous neighboring polar nanoregions is the main mechanism that explains their very high electromechanical response [10,11].

Ecological requirements have stimulated a search for lead-free alternatives to the aforementioned piezoelectric materials. Solid solutions between well-known ferroelectrics and antiferroelectrics such as BaTiO_3_, NaNbO_3_, KNbO_3_, Na_1/2_Bi_1/2_TiO_3_ and K_1/2_Bi_1/2_TiO_3_ were studied in this respect [12]. Some of the systems, e.g., K_1/2_Bi_1/2_TiO_3_–BaTiO_3_, which were found to be promising, are under intense study to enhance the electromechanical characteristics and adjust them to the operating range as well as to make the ceramics production process simpler and easily reproducible. At the same time, many efforts are focused on discovering new lead-free perovskite systems with MPB. Attempting to reveal new compositions, researchers explore multicomponent systems, in which one or more end members do not have a stable modification at ambient pressure (e.g., BiScO_3_ [13], BiMnO_3_ [14] and BiCrO_3_ [15]). In some cases, the compositions corresponding to MPB in such systems can be prepared using conventional routes; the majority of cases require the application of high-pressure synthesis. Novel solid solution systems with MPB stabilized under high pressure have been reported [16,17,18,19,20]. In the MPB ranges of the BiCo_1-x_FexO_3_ [17] and Na_1/2_Bi_1/2_V_1-*x*_Ti*_x_*O_3_ [20] systems, a monoclinic phase with polarization rotation similar to that observed in PZT was detected.

In the (1-*x*)BiMg_0.5_Ti_0.5_O_3_–*x*BiZn_0.5_Ti_0.5_O_3_ system [(1-*x*)BMT–*x*BZT], in which the orthorhombic BMT is a structural analogue of PbZrO_3_ [21], while the tetragonal BZT is isostructural to PbTiO_3_ [22], neither the end members nor their solid solutions can be obtained in bulk form through conventional routes. The as-prepared (unannealed) compositions with a relative BZT content *x* < 0.75 are orthorhombic (space group *Pnnm*), while those with a BZT content above this value are tetragonal (*P*4*mm*). In the solution with *x* = 0.75, both phases coexist, forming an MPB [23].

A phenomenon of annealing-stimulated irreversible transformations of the high-pressure stabilized phases (conversion polymorphism) [24] has recently been revealed. It was shown that the pattern of the phase diagram of the high-pressure prepared complex perovskite compositions, which demonstrates the effect of conversion polymorphism, depends on the maximum annealing temperature. This feature can be used to design new materials with MPB since by means of controlled annealing, materials with different combinations of the perovskite phases can be obtained.

In this paper, we study the effect of annealing on the state of high-pressure sintered (1-*x*)BMT–*x*BZT ceramics with compositions around the MPB. We report on the in situ temperature X-ray diffraction study and the Piezoresponse Force Microscopy measurements of the (1-*x*)BMT–*x*BZT ceramics and demonstrate the annealing-stimulated irreversible changes in the structural phase diagram and in the polar domain structure of the ceramics. The phase diagram of the annealed BMT–BZT solid solutions is analyzed in comparison with that of PZT.

## 2. Materials and Methods

The (1-*x*)BMT–*x*BZT ceramics were produced using high-pressure high-temperature technique 6 GPa and 1470–1570 K. Synthesis time did not exceed 10 min. Details of the precursor preparation and the high-pressure synthesis can be found in Ref. [25].

The microstructural characterization of the ceramics before and after annealing was performed using a Hitachi S-4100 scanning electron microscope (SEM) (Hitachi, Chiyoda, Japan) operated at 25 kV.

To characterize the phase content and the crystal structure of the samples via X-ray diffraction (XRD), the as-prepared ceramics were reduced into fine powders. XRD measurements were performed using a PANalytical X’Pert Powder diffractometer (Malvern Panalytical, Malvern, UK) (Ni filtered Cu Kα radiation, an exposition of about 2 s per 0.02° step over a 2-theta range of 15–85°). In situ temperature-dependent XRD measurements were carried out in an Anton Paar HTK 16 N chamber between 300 and 820–1020 K upon both heating and cooling. The samples (in powder form) were kept for 30 min at each temperature point before the XRD data collection to ensure equilibrium condition. The obtained XRD data were refined by the Rietveld method using the FULLPROF 7.60 package (Rennes-Grenoble, France) [26].

Piezoresponse force microscopy (PFM) studies were performed using a commercial scanning probe microscope MFP-3D (Asylum Research, Oxford Instruments, Santa Barbara, CA, USA). Pt coated cantilevers Multi75E-G (Budget Sensors, Sofia, Bulgaria) with a spring constant of 3 N/m were used. The PFM measurements were conducted at probing voltage with an amplitude *U_ac_* = 5 V and a frequency close to the contact resonance frequency of the cantilevers (*f*~380 kHz and 700 kHz for the vertical and lateral PFM signals, respectively). The PFM images were analyzed using the Gwyddion 2.45 software (Gwyddion, Brno, Czech Republic) [27].

## 3. Results and Discussion

All the perovskite compositions of the (1-*x*)BMT–*x*BZT system were prepared using high-pressure synthesis [23]; the attempts to produce these materials via conventional ceramics methods resulted in the formation of a mixture of non-perovskite phases [25]. To estimate the thermal stability of the obtained materials at ambient pressure, high-pressure synthesized samples were thermally treated (annealed) in air and their phase transformation(s) were controlled in situ using a high-temperature XRD chamber. The maximum temperature was 1020 K. The thermal stability limit was defined as the temperature which is 50 K lower than that when the diffraction reflections of a non-perovskite phase (Bi_4_Ti_3_O_12_) appear or start to increase. The stability limit of perovskite phases in the (1-*x*)BMT–*x*BZT system was found to depend on the BZT content and decreases as *x* increases (Table 1). Thus, the perovskite (1-*x*)BMT–*x*BZT phases are all metastable regardless of their crystal symmetry.

SEM study has revealed no regular change in morphology of the ceramics dependent on their composition. The microstructure of a 0.30BMT–0.70BZT sample shown in Figure 1a is typical of all the ceramics under study. Annealing of the samples at temperatures below the thermal stability limit of their perovskite phase(s) was found to lead to no visible change of their microstructure (cf: Figure 1b). However, even a relatively short (about 20 min) thermal treatment at higher temperatures resulted in a drastic modification of the sample morphology (Figure 1c). The phase content of the ceramics after the decomposition of the perovskite phase was essentially similar to that of the synthesis product of samples of the same composition at ambient pressure.

An in situ temperature XRD study of the (1-*x*)BMT–*x*BZT solid solutions carried out at temperatures below the thermal stability limits revealed irreversible structural transitions of the perovskite phases over a wide compositional range. The orthorhombic structure of the as-prepared (unannealed) samples with 0.60 ≤ *x* ≤ 0.75 transforms into the tetragonal structure upon heating and this structure remains upon cooling down to room temperature. Figure 2 shows the XRD patterns that demonstrate such a transformation in a sample with *x* = 0.70.

For all the solid solutions from the aforementioned composition range, the transition starts at about 470 K and completes at 520–570 K. The orthorhombic structure of the unannealed samples from the range of 0.20 ≤ *x* ≤ 0.60 transforms irreversibly to the rhombohedral structure. For these compositions, the transition initiation temperature is close to 420 K, and the phase coexistence range is about 100 K. The characteristic evolution of the XRD pattern is shown in Figure 3 for the case of a solid solution with *x* = 0.40.

The (1-*x*)BMT–*x*BZT solid solutions with *x* ≤ 0.10 keep the orthorhombic structure over the annealing.

Based on the results of in situ temperature XRD study, the phase diagrams of the as-prepared (1-*x*)BMT–*x*BZT perovskite compositions upon heating (Figure 4a) and cooling (Figure 4b) were plotted.

The second and all subsequent heating/cooling thermal cycles of the annealed (1-*x*)BMT–*x*BZT samples with a maximum temperature not exceeding their temperature stability limit have demonstrated neither reversible nor irreversible structural transformation. The phase diagram of the annealed samples remained the same upon both heating and cooling (Figure 4b).

The crystal structure of the observed perovskite phases was successfully refined using the orthorhombic *Pnnm* space group with the √2*a_p_* × 2*a_p_* × 2√2*a_p_* superstructure (where *a_p_* is the pseudocubic primitive perovskite unit-cell parameter), the rhombohedral *R*3*c* group (√2*a_p_* × √2*a_p_* × 2√3*a_p_*) and the tetragonal *P*4*mm* group (*a_p_* × *a_p_* × *a_p_*). As pointed out in Ref. [21], due to close to tetragonal metric of the pseudocubic perovskite cell, it is difficult to establish the right space group unambiguously for the orthorhombic phase, and the symmetry might be *Pnma* instead of *Pnnm*. The secondary phase identified as the orthorhombic Bi_4_Ti_3_O_12_ [28] (space group *B*2*cb*), which was found to be present as 3–5 mol. % impurity [23], was also taken into consideration.

Figure 5 demonstrates the compositional dependence of the normalized perovskite unit-cell volume (*V_p_* = *V*/*Z*) of the (1-*x*)BMT–*x*BZT solid solutions before and after annealing. It is clearly seen that except for the case of the BMT end member (*x* = 0), in the ranges where no transformation occurred (0 < *x* ≤ 0.1 and 0.75 ≤ *x* ≤ 1) as well as where the orthorhombic phase transformed into the tetragonal one (0.60 ≤ *x* < 0.75), the annealing resulted in an increase of the *V_p_* value.

It is interesting that the *V_p_(x)* dependence in the ranges of the annealed orthorhombic and the annealed tetragonal phases obeys well the linear Vegard’s law and increases as *x* is increased (solid line in Figure 5). At the same time, in the compositional range between *x* = 0.20 and 0.60, where the irreversible orthorhombic-to-rhombohedral transition occurred, the unit-cell volume changes slowly and exhibits the trend to decrease with increasing *x*. Such an unusual behavior suggests the dominant role of the covalent bonds over the electrostatic interactions and the local crystal structure distortions in this range of the system that can result in interesting combinations of polar and elastic ferroic orders.

The new location of the MPB in the (1-*x*)BMT–*x*BZT system, where the rhombohedral and the tetragonal phases coexist, is at *x* = 0.60. Thus, annealing resulted not only in the orthorhombic-to-rhombohedral transformation in the vicinity of the MPB but also in a compositional shift of the boundary by 15 mol. % in the direction of a lower BZT content. A relative jump in the *V_p_* value upon crossing the MPB with increasing *x* was estimated to be about 3% as compared with 5% observed in the as-prepared (1-*x*)BMT–*x*BZT solid solutions.

The refined lattice parameters of the annealed samples with the *Pnnm* phase and the *R*3*c* phase were recalculated to the values of the primitive perovskite lattice parameters: *a_p_* = *b_p_* ≠ *c_p_*, *α_p_* = *β_p_* = 90° ≠ *γ_p_* and *a_p_* = *b_p_* = *c_p_*, *α_p_* = *β_p_* = *γ_p_* ≠ 90° for the orthorhombic and the rhombohedral compositions, respectively [29]. The obtained values were plotted together with the *a_p_(x)* and *c_p_(x)* parameters of the tetragonal compositions for comparison. Figure 6 shows the compositional dependences of the primitive perovskite unit-cell parameters of the (1-*x*)BMT–*x*BZT solid solutions before and after the annealing.

The compositional dependence of the pseudocubic primitive perovskite unit-cell parameters is essentially similar to that of PZT [1].

The change in symmetry should be reflected in changes of the functional properties of the (1-*x*)BMT–*x*BZT ceramics, namely in the onset of the ferroelectric behavior. Unfortunately, small and brittle samples obtained after the high-pressure synthesis are unsuitable for macroscopic ferroelectric measurements. Nevertheless, ferroelectricity in such samples can be probed at the local scale using piezoresponse force microscopy [30]. Recently, we reported on the domain structure and local ferroelectric properties in the as-prepared (1-*x*)BMT–*x*BZT ceramics with compositions around the MPB that was detected at *x* = 0.75 [23]. In this study we focus on the compositions from the BMT-side to the MPB, with *x* = 0.55–0.70. Figure 7a–c and Figure 8a–c show the PFM images of as-prepared ceramics with *x* = 0.55 and 0.65. Vertical PFM images for other compositions are compared in Appendix A. The images are presented in false colors, where the dark and bright contrast correspond to regions with small and large piezoresponse, respectively. One can see that, for the as prepared samples, most of the area shows a negligible piezoresponse in agreement with the non-polar macroscopic symmetry of the orthorhombic phase. However, some piezoactive regions are observed. These regions can be attributed to ferroelectric domains of the tetragonal phase similar to one in the compositions from the BZT side. The area occupied by these piezoactive domains increases in the composition with larger *x*, i.e., upon approaching the MPB.

Annealing results in drastic changes in the PFM images. Figure 7d–f and Figure 8d–f show PFM images of the same areas of 0.45BMT–0.55BZT and 0.35BMT–0.65BZT ceramics taken after ex situ annealing at 623 K for 1 h.

It can be seen that regions of the strong piezoresponse increased drastically and occupied almost the whole studied area. The statistical analysis showed that the relative area occupied by the ferroelectric regions increases approximately by a factor of 6 or 7. This observation confirms that the annealing resulted in a transformation of the initially non-polar material into the polar one. We observed both growth of already existing ferroelectric domains and the appearance of new ones. This can be clearly seen in the magnified PFM images (Appendix A). It has to be noted that the domains in the annealed ceramics with *x* = 0.55 and 0.65 have different morphology. While in the 0.35BMT–0.65BZT ceramics, fine domains with irregular shape are observed after annealing, in the annealed 0.45BMT–0.55BZT ceramics, the domains are larger and with more regular orientation. This may be due to the different crystalline symmetry of the annealed state, tetragonal and rhombohedral in the former and later cases, respectively.

## 4. Conclusions

The perovskite (1-*x*)BiMg_0.5_Ti_0.5_O_3_–*x*BiZn_0.5_Ti_0.5_O_3_ system is an interesting example of conversion polymorphism. The solid solutions of this system, prepared by high-pressure synthesis, have a phase diagram with an MPB between the non-polar orthorhombic and polar tetragonal phases at *x* = 0.75. Annealing at temperatures above 573 K leads both to an irreversible transition of compositions with 0.2 < *x* < 0.6 to the polar rhombohedral state and to a shift of the morphotropic phase boundary with the tetragonal state towards the BMT-side (*x* = 0.6). The transformation from the non-ferroelectric to the ferroelectric state with a strong piezoelectric signal is confirmed by PFM measurements. The phase diagram of the annealed ceramics is very similar to that of the famous lead zirconate titanate ceramics, which dominate the market for piezoelectric applications but need to be replaced by environmentally friendly lead-free systems. The results obtained allow us to state that the BMT–BZT system is promising for the creation of materials with high electromechanical and dielectric characteristics.

## Figures and Tables

**Figure 1 materials-15-06998-f001:**
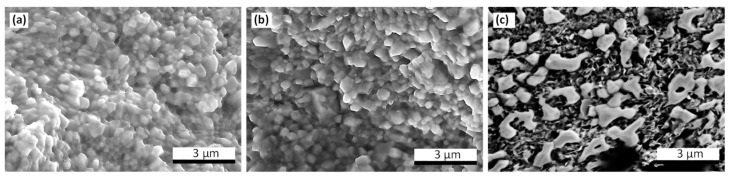
SEM images of the fractured surfaces of the (1-*x*)BMT–*x*BZT sample (*x* = 0.70) synthesized under high pressure: (**a**) as prepared, (**b**) annealed at 770 K and (**c**) annealed at 970 K.

**Figure 2 materials-15-06998-f002:**
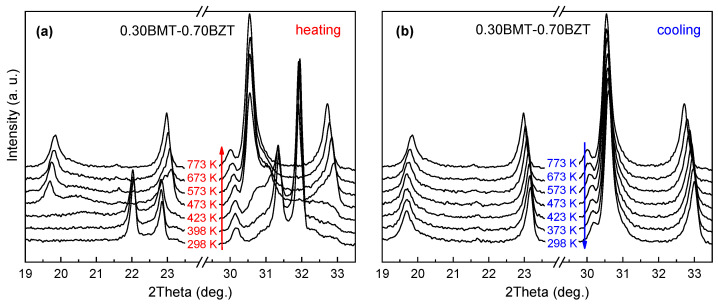
The most representative ranges of the XRD patterns of the as-prepared (1-*x*)BMT–*x*BZT sample (*x* = 0.70) at the first thermal cycle: (**a**) upon heating to 773 K and (**b**) upon cooling to room temperature. The structural phases at room temperature before and after the annealing are the orthorhombic *Pnnm* and the tetragonal *P*4*mm*, respectively.

**Figure 3 materials-15-06998-f003:**
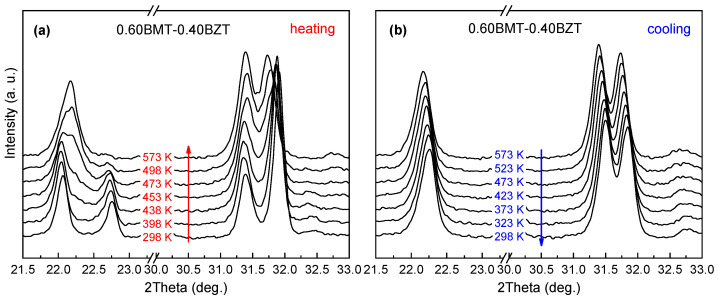
The most representative ranges of the XRD patterns of the as-prepared (1-*x*)BMT–*x*BZT sample (*x* = 0.40) at the first thermal cycle: (**a**) upon heating to 773 K and (**b**) upon cooling to room temperature. The structural phases at room temperature before and after the annealing are the orthorhombic *Pnnm* and the tetragonal *R*3*c*, respectively.

**Figure 4 materials-15-06998-f004:**
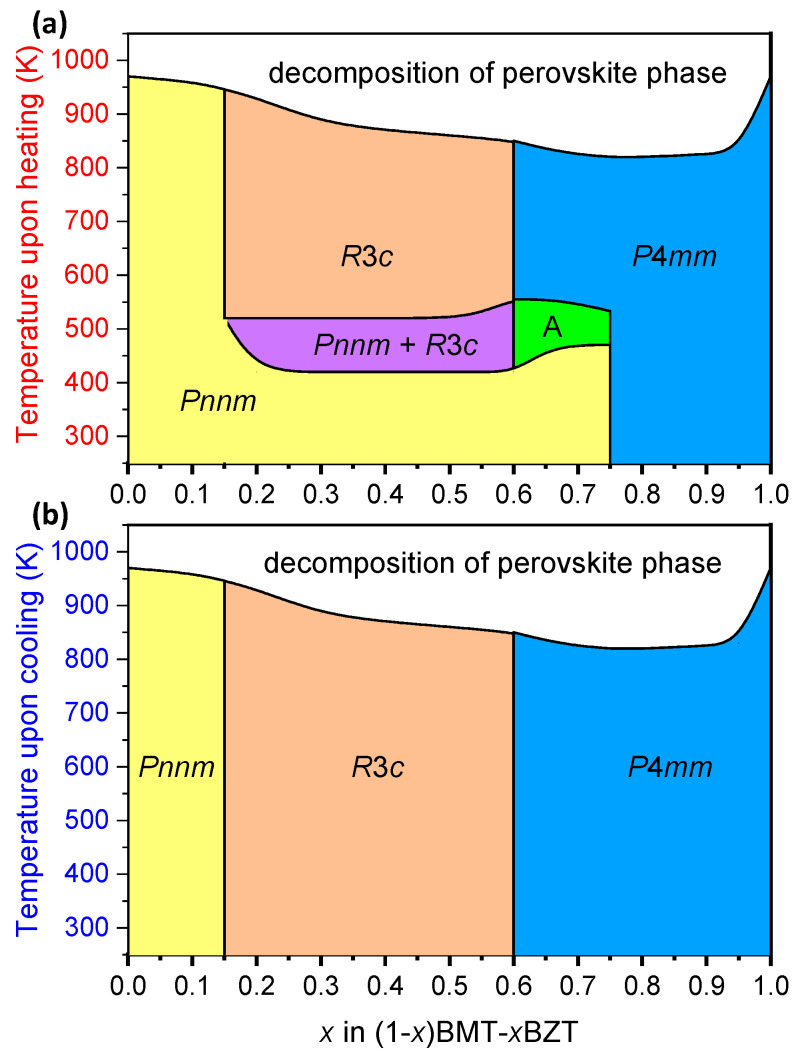
State diagram of the (1-*x*)BMT–*x*BZT perovskite phases upon heating to their temperature stability limit (**a**) and upon subsequent cooling (**b**). The area marked as “A” corresponds to the *Pnnm* + *P*4*mm* phase mixture.

**Figure 5 materials-15-06998-f005:**
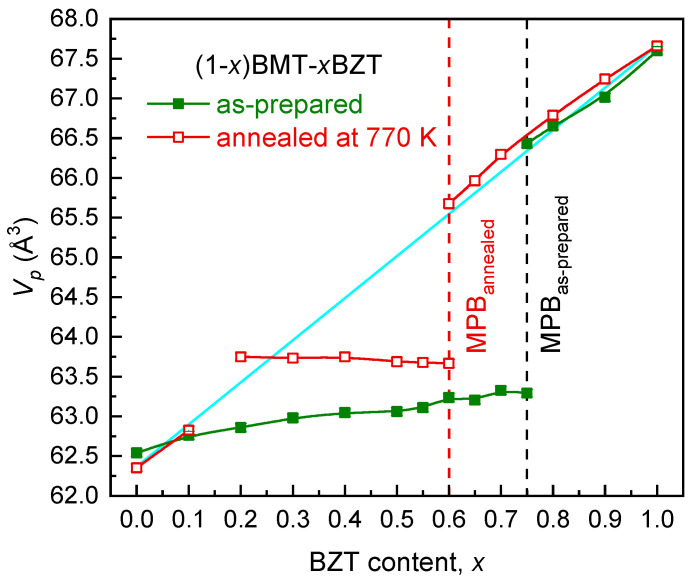
The normalized unit-cell volume of the (1-*x*)BMT–*x*BZT perovskite phases before and after annealing as a function of the BZT content (*x*). The error bars are smaller than the symbols. The dashed lines indicate the compositions corresponding to the morphotropic phase boundary (MPB) before and after annealing. The solid straight line is a reference to the Vegard’s law.

**Figure 6 materials-15-06998-f006:**
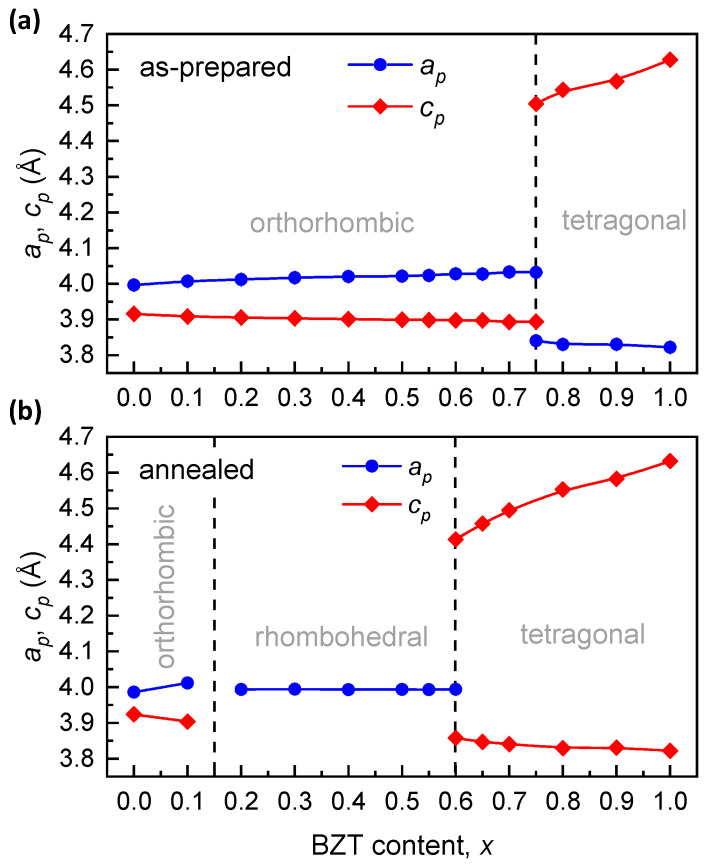
The primitive perovskite cell parameters of the (1-*x*)BMT–*x*BZT perovskite phases as a function of the BZT content (*x*) before (**a**) and after (**b**) annealing with the borders of the different phase ranges indicated.

**Figure 7 materials-15-06998-f007:**
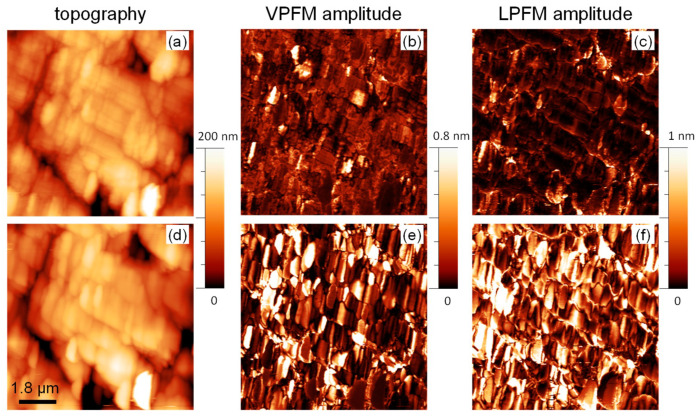
Topography (**a**,**d**), vertical PFM amplitude (**b**,**e**), and lateral PFM amplitude (**c**,**f**) images of the 0.45BMT–0.55BZT ceramics before (**a**–**c**) and after annealing (**d**–**f**).

**Figure 8 materials-15-06998-f008:**
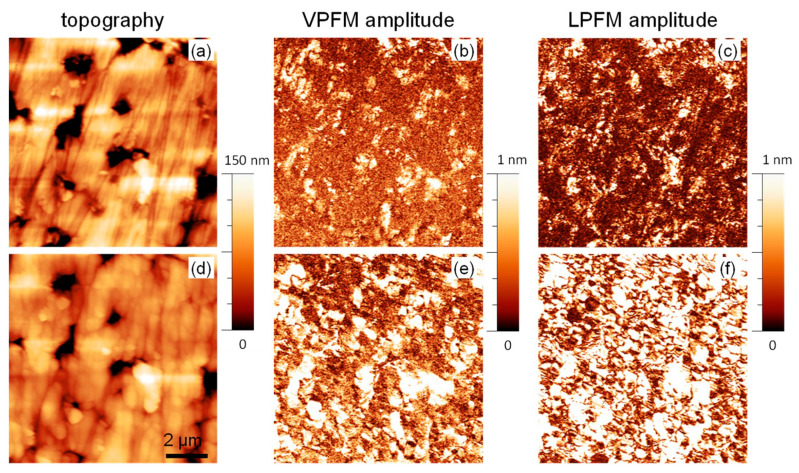
Topography (**a**,**d**), vertical PFM amplitude (**b**,**e**), and lateral PFM amplitude (**c**,**f**) images of the 0.35BMT–0.65BZT ceramics before (**a**–**c**) and after annealing (**d**–**f**).

**Table 1 materials-15-06998-t001:** The thermal stability limit of the high-pressure synthesized perovskite phases in the (1-*x*)BMT–*x*BZT system.

The Composition Range	The Thermal Stability Limit, K
*x* ≤ 0.20	970
0.20 *< x* ≤ 0.65	870
*x* > 0.65	820

## Data Availability

The data presented in this study are available on request from the corresponding author.

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
