# Peer review of "Annealing-Dependent Morphotropic Phase Boundary in the BiMg_0.5_Ti_0.5_O_3_–BiZn_0.5_Ti_0.5_O_3_ Perovskite System"

_materials, 2022, doi:10.3390/ma15196998_

Round 1
Reviewer 1 Report
1. Authors have to elaborate on the impact of the environment on the synthesized materials.
2. A more in-depth study is required to understand the behavior of the synthesized material.
Author Response
- Authors have to elaborate on the impact of the environment on the synthesized materials.
Answer:
It is not very clear what the reviewer means. At ambient conditions, the synthesized materials are stable and demonstrate no tendency to degradation. The stability of these materials in various atmospheres or in the presence of certain chemically active substances may be the subject of a particular study, which is out of topic of the presented work.
- A more in-depth study is required to understand the behavior of the synthesized material.
Answer:
Following the comment, we have added information regarding the behavior of the synthesized materials during subsequent annealing cycles and a phase diagram (new Figure 4)
“The second and all subsequent heating/cooling thermal cycles of the annealed (1-x)BMT–xBZT samples with a maximum temperature not exceeding their temperature stability limit, demonstrated neither reversible nor reversible structural transformation. The phase diagram of the annealed samples remained the same upon both heating and cooling (Figure 4b).”
Reviewer 2 Report
(a) The manuscript should be thoroughly polished in language.
(b) Experimental details in sample preparation should be provided.
Author Response
(a) The manuscript should be thoroughly polished in language.
Answer:
English grammar and style have been polished
(b) Experimental details in sample preparation should be provided.
Answer:
We have added sample preparation details to Part 2 – Materials and Methods
“The (1-x)BMT-xBZT ceramics were produced using high-pressure high-temperature technique at 6 GPa and 1470-1570 K. The synthesis time did not exceed 10 min. Details on the preparation of the precursor preparation can be found in Ref. [25].”
Reviewer 3 Report
The manuscript describes the phase transition in BMT-BZT composite observed by in-situ X-ray diffraction (XRD) and piezoresponse force microscopy (PFM). The composite irreversible transformed to the polar rhombohedral phase when BZT content is between 0.2 to 0.6, and to the tetragonal phase when BZT content > 0.6. The results are suitable for publication in the journal. Some comments below should be considered for better presentation.
1. The manuscript presents one-cycle heating-cooling. What is about multi-cycles thermal load? It is useful to evaluate the thermal stability of the composite.
2. The conclusion mentions mechanical and dielectric properties. Evaluations on such properties are recommended.
Author Response
We are grateful to the reviewer for his (her) overall positive assessment of our manuscript.
- The manuscript presents one-cycle heating-cooling. What is about multi-cycles thermal load? It is useful to evaluate the thermal stability of the composite
Answer:
After the first annealing, the subsequent heating-cooling cycles resulted in no structural/phase change of the materials regardless of the composition. We added the following text:
“The second and all subsequent heating/cooling thermal cycles of the annealed (1-x)BMT–xBZT samples with a maximum temperature not exceeding their temperature stability limit have demonstrated neither reversible nor reversible structural transformation. The phase diagram of the annealed samples remained the same upon both heat-ing and cooling (Figure 4b).”
- The conclusion mentions mechanical and dielectric properties. Evaluations on such properties are recommended.
Answer:
As mentioned in the manuscript, ceramics prepared by high-pressure synthesis are small and brittle, and therefore unsuitable for macroscopic dielectric and electromechanical measurements. This was the main motivation for characterizing their ferroelectric and electromechanical properties using a local probe method such as PFM. We are currently looking for a synthesis route capable of producing samples suitable for macroscopic measurements.
Reviewer 4 Report
The mentioned manuscript is subject to corrections.
1. The title is to be: Annealing temperature effect on Morphotropic Phase Boundary in the
3 BiMg0.5Ti0.5O3 - BiZn0.5Ti0.5O3 Perovskite oxides
2. Polish the English language
3. Section 1. To improve the manuscript quality for perovskite, cite: Journal of the American Ceramic Society 104 (2021) 6508-6520; Results in Physics 42 (2022) 105977;
4. Section 1. Highlight the objective of the work
5. Correlate structural & morphological properties
6. State the main findings in the conclusions.
7. Some refs are out of date (more than 10 years). Update!
8. Remove less significant and unrelated or less related references and ensure that all the references are cited and arranged sequentially as required by the journal.
Author Response
We are gaterful to the reviewer for his (her) reccomendation.
- The title is to be: Annealing temperature effect on Morphotropic Phase Boundary in the BiMg0.5Ti0.5O3 - BiZn0.5Ti0.5O3 Perovskite oxides
Answer:
We are grateful to the reviewer for the suggestion. However, we believe that our original title correctly reflects the observed effect. It is also shorter.
- Polish the English language
Answer:
English grammar and style have been polished
3.Section 1. To improve the manuscript quality for perovskite, cite: Journal of the American Ceramic Society 104 (2021) 6508-6520; Results in Physics 42 (2022) 105977;
Answer: Thanks to the reviewer for the suggestion. We have read the articles recommended by the reviewer. However, their subject is far from the topic of our study. Therefore, we decided not to cite them.
- Section 1. Highlight the objective of the work
Answer:
The objective of the work is now highlighted in the Introduction (the last paragraph):“In this paper, we study the effect of annealing on the state of high-pressure sintered (1-x)BMT-xBZT ceramics with compositions around the MPB.”
- Correlate structural & morphological properties
Answer: It is stated in the manuscript (lines 127-128) that the SEM study did not reveal a regular change in the morphology of the ceramics depending on their composition. It is easy to understand from this statement that in our case there are no essential reasons to correlate structural and morphological properties of the ceramics.
- State the main findings in the conclusions.
Answer:
Main findings are stated in the Conclusions.
- Some refs are out of date (more than 10 years). Update!
Answer:
Many old works, even of the last century are still actual. Moreover, we prefer to cite original works.
- Remove less significant and unrelated or less related references and ensure that all the references are cited and arranged sequentially as required by the journal.
Answer:
The list of citations has been prepared in accordance with the rules of the Materials.
Round 2
Reviewer 2 Report
The manuscript is recommended for publication as it is.
Author Response
We are grateful to the reviewer for the recommendation.
Reviewer 4 Report
The authors did not revise the manuscript based on my previous all comments. Also, the authors did not take my comments seriously. So, at this stage, I can not accept the manuscript for publication. So, for the consideration of this paper for possible publication, the author must revise the manuscript based on my previous comments along with the following comments:
(1) Literature review needs to include several recent, relevant publications (high impact) highlighting their key findings. The current version only discussed general aspects while the review of each from several papers is necessary. You may provide a review summary table consisting of a column for the comments or key conclusions.
(2) Enhance the objective and novelty of the work in the introduction section.
(3) The title is to be: Annealing temperature effect on Morphotropic Phase Boundary in the BiMg0.5Ti0.5O3 - BiZn0.5Ti0.5O3 Perovskite oxides
(4) Section 1. To improve the manuscript quality for perovskite, cite: Journal of the American Ceramic Society 104 (2021) 6508-6520; Results in Physics 42 (2022) 105977.
(5) Correlate structural & morphological properties
(6) Some refs are out of date (more than 10 years). Update them and replace them with recent publications.
(7) Similarity is 34% which is not acceptable. In sections 1 and 2, the author directly copied text from other sources. Reduce similarity.
Authors are suggested to take all the above comments seriously during revision submission. Otherwise, I can not accept the paper for publication.
Author Response
The reviewer repeated his comments. These comments concern mainly the style of the writing, and not the subject of the manuscript, Below we give a point by point response.
- “Literature review needs to include several recent, relevant publications (high impact) highlighting their key findings. The current version only discussed general aspects while the review of each from several papers is necessary. You may provide a review summary table consisting of a column for the comments or key conclusions.”
This is a very general comment. The reviewer did not explicitly indicate which recent relevant publications should be included. The style of writing a literature review is individual. Materials and other MDPI journals do not require a specific literature review format. The other three reviewers had no objection to the style of the literature review. Therefore, we consider this comment as subjective and cannot agree with it.
- “Enhance the objective and novelty of the work in the introduction section.”
The objectives and novelty of the work are clearly formulated in the last two paragraph of the Introduction and in the conclusions.
In Introduction
“A phenomenon of annealing-stimulated irreversible transformations of the high-pressure stabilized phases (conversion polymorphism) [24] has recently been revealed. It was shown that the pattern of the phase diagram of the high-pressure pre-pared complex perovskite compositions, which demonstrates the effect of conversion polymorphism, depends on the maximum annealing temperature. This feature can be used to design new materials with MPB since by means of controlled annealing, materials with different combinations of the perovskite phases can be obtained.
In this paper, we study the effect of annealing on the state of high-pressure sintered (1-x)BMT-xBZT ceramics with compositions around the MPB. We report on the in situ temperature X-ray diffraction study and the Piezoresponse Force Microscopy measurements of the (1-x)BMT-xBZT ceramics and demonstrate the annealing-stimulated irreversible changes in the structural phase diagram and in the polar domain structure of the ceramics. The phase diagram of the annealed BMT-BZT solid solutions is analysed in comparison with that of PZT.”
In Conclusion
“The phase diagram of the annealed ceramics is very similar to that of the famous lead zirconate titanate ceramics, which dominate the market for piezoelectric applications but need to be replaced by environmentally friendly lead-free systems. The results obtained allow us to state that the BMT-BZT system is promising for the creation of materials with high electromechanical and dielectric characteristics.”
We don’t see the need for further enhancement of these clear statements.
- The title is to be: Annealing temperature effect on Morphotropic Phase Boundary in the BiMg0.5Ti0.5O3 - BiZn0.5Ti0.5O3 Perovskite oxides
The choice of the title is the right of the authors. The reviewer can recommend, but cannot demand this. In our previous reply, we explained why we decided to keep the original title. “We are grateful to the reviewer for the suggestion. However, we believe that our original title correctly reflects the observed effect. It is also shorter.“
- To improve the manuscript quality for perovskite, cite: Journal of the American Ceramic Society 104 (2021) 6508-6520; Results in Physics 42 (2022) 105977.
The reviewer has not justified why these papers have to be cited. The first paper is by M. Khalid Hossain et al. “Hydrogen isotope dissolution and release behavior in Y-doped BaCeO3” and the second paper is by M. H. K. Rubel et al. “First-principles calculations to investigate structural, elastic, electronic, thermodynamic, and thermoelectric properties of CaPd3B4O12 (B = Ti, V) perovskites”. Both of these papers discuss neither the phenomenon of conversion polymorphism nor the properties of piezoelectric/ferroelectric ceramics from a morphotropic phase boundary. That is, they are completely irrelevant to the subject of our study. We don’t understand why the reviewer have insisted on citing them and not on any other of the thousands of published papers on materials with the perovskite structure.
- Correlate structural & morphological properties.
We already answered this comment. “It is stated in the manuscript (lines 127-128) that the SEM study did not reveal a regular change in the morphology of the ceramics depending on their composition. It is easy to understand from this statement that in our case there is no direct correlation between structural and morphological properties of the ceramics.
- Some refs are out of date (more than 10 years). Update them and replace them with recent publications.
The year of publication is not an argument for not citing an article. The reviewer must justify which citation is outdated and why. Many old works, even from the last century, are still relevant. Moreover, we prefer to cite original and seminal works.
- Similarity is 34% which is not acceptable. In sections 1 and 2, the author directly copied text from other sources. Reduce similarity.
This is a new comment, which was absent in the previous report. Section 2 gives a standard description of the experimental methods. It is expected to bear some similarity to our previous publications. In other cases, the reviewer has to provide evidences that the text was directly copied.